# Identification of Novel Substrates for cGMP Dependent Protein Kinase (PKG) through Kinase Activity Profiling to Understand Its Putative Role in Inherited Retinal Degeneration

**DOI:** 10.3390/ijms22031180

**Published:** 2021-01-25

**Authors:** Akanksha Roy, John Groten, Valeria Marigo, Tushar Tomar, Riet Hilhorst

**Affiliations:** 1Division of Toxicology, Wageningen University and Research, Stippeneng 4, 6708 WE Wageningen, The Netherlands; jgroten@pamgene.com; 2PamGene International B.V., P.O. Box 1345, 5200 BJ ‘s-Hertogenbosch, The Netherlands; TTomar@pamgene.com (T.T.); rhilhorst@pamgene.com (R.H.); 3Department of Life Sciences, University of Modena and Reggio Emilia, via Campi, 287, 41125 Modena, Italy; vmarigo@unimore.it; 4Center for Neuroscience and Neurotechnology, University of Modena and Reggio Emilia, via Campi, 287, 41125 Modena, Italy

**Keywords:** PKG, PKA, cGMP, cAMP, 661W, retinal degeneration, substrate identification, peptide microarray

## Abstract

Inherited retinal degenerative diseases (IRDs), which ultimately lead to photoreceptor cell death, are characterized by high genetic heterogeneity. Many IRD-associated genetic defects affect 3′,5′-cyclic guanosine monophosphate (cGMP) levels. cGMP-dependent protein kinases (PKGI and PKGII) have emerged as novel targets, and their inhibition has shown functional protection in IRDs. The development of such novel neuroprotective compounds warrants a better understanding of the pathways downstream of PKGs that lead to photoreceptor degeneration. Here, we used human recombinant PKGs in combination with PKG activity modulators (cGMP, 3′,5′-cyclic adenosine monophosphate (cAMP), PKG activator, and PKG inhibitors) on a multiplex peptide microarray to identify substrates for PKGI and PKGII. In addition, we applied this technology in combination with PKG modulators to monitor kinase activity in a complex cell system, i.e. the retinal cell line 661W, which is used as a model system for IRDs. The high-throughput method allowed quick identification of *bona fide* substrates for PKGI and PKGII. The response to PKG modulators helped us to identify, in addition to ten known substrates, about 50 novel substrates for PKGI and/or PKGII which are either specific for one enzyme or common to both. Interestingly, both PKGs are able to phosphorylate the regulatory subunit of PKA, whereas only PKGII can phosphorylate the catalytic subunit of PKA. In 661W cells, the results suggest that PKG activators cause minor activation of PKG, but a prominent increase in the activity of cAMP-dependent protein kinase (PKA). However, the literature suggests an important role for PKG in IRDs. This conflicting information could be reconciled by cross-talk between PKG and PKA in the retinal cells. This must be explored further to elucidate the role of PKGs in IRDs.

## 1. Introduction

Inherited retinal degenerative diseases (IRDs) include a group of diseases that lead to severe vision impairment and blindness at a young age due to mutations affecting the functioning of the retina. Retinitis pigmentosa is one of the most prominent and most heterogenous IRD, with over 65 defective genes identified in the autosomal dominant, recessive, and X-linked forms of the disease [1]. During the onset of retinitis pigmentosa, rod photoreceptors, responsible for vision under dim light, are primarily affected. The disease manifests itself with the patients experiencing night vision problems and tunnel vision due to a loss of rod cells that are present at the periphery of the retina. With the progression of the disease, cone photoreceptors are also degenerated, leading to complete blindness [2].

The phototransduction pathway in retina is mediated by fluctuations in cGMP levels in the photoreceptors [2]. In the dark, high cGMP levels open cyclic nucleotide gated (CNG) ion channels. These non-specific channels lead to an influx of Na^+^ and Ca^2+^ into the photoreceptors. At the same time, K^+^ and Ca^2+^ are continuously extruded via a Na^+^/K^+^/Ca^2+^ exchanger. The influx and efflux of ions via these two types of channels lead to generation of a dark current. Light disrupts the generation of the dark current. The absorption of photons induces conformational changes in rhodopsin (the visual pigment present in rod photoreceptors) which, through a series of steps, leads to the activation of phosphodiesterase-6 (PDE6). PDE6 hydrolyzes cGMP and prompts the closure of CNG channels, resulting in hyperpolarization due to the efflux of Ca^2+^ and K^+^ by the Na^+^/K^+^/Ca^2+^ exchanger, which sends a signal to the neuronal cells and ultimately to the brain for visual processing [2,3].

Mutations in the rod-specific PDE6 subunits α and β have been linked to high intracellular levels of cGMP, as cGMP is not hydrolyzed. For photoreceptor degeneration, three times higher cGMP levels have been reported in murine retinas with a PDE6 mutation [4,5]. Mutations in other genes encoding for rhodopsin, CNG ion channels, aryl hydrocarbon interacting protein-like 1 or photoreceptor guanylyl cyclase have also been identified to cause excessive cGMP accumulation in the photoreceptors [2,6]. cGMP opens the CNG ion channels and activates PKGs by binding to their regulatory sites. PKGs phosphorylate components of ion channels, G-proteins, and cytoskeleton proteins that regulate neuronal, cardiovascular, and intestinal functions [7]. Application of PKG inhibitors in vivo in retinas of the *rd1* mouse model resulted in photoreceptor protection [8]. Recently, it was demonstrated that in vivo application of a PKG inhibitor with a drug delivery system counteracted photoreceptor degeneration and preserved retina function in three IRD mouse models [9]. In addition, it has been shown in a CNG channel loss-of function mouse model that knocking out of *prkg1* encoding for PKGI leads to sustained rod cell survival [10]. The correlation of PKG activation with cell death is corroborated by studies showing induction of apoptosis with PKG activation to stop tumor progression in colon cancers [11,12,13], breast cancers [14], ovarian cancers [15] and melanoma [16]. However, the signaling routes downstream of PKGI and PKGII that lead to cell death have not been identified yet. Since in this heterogeneous disease, many mutations convene in a common cGMP-driven PKG signaling route, targeting PKGs could provide a common treatment amenable for a larger group of patients. Therefore, identification of targets for PKGI and PKGII as well as their downstream signaling pathways might help to develop novel treatments for rare diseases such as IRDs, but also for various cancer types.

The aim of this study was to identify high-quality substrates for PKGI and PKGII, and use this information to identify critical PKG targets in the retina. Since a kinase has multiple substrates, we used a peptide microarray with 142 serine/threonine containing peptides to determine preferred PKGI or PKGII substrates. PKG activity was modulated by known PKG specific inhibitors and activators that bind to the regulatory sites of cGMP-dependent protein kinases to identify novel and bona fide substrates for PKGI and PKGII. We then applied the knowledge on modulation of PKG activity to investigate the role of endogenous PKGI and PKGII in 661W retinal cells that are also used as model for retinal degeneration [17,18]. We show that in a complex cellular environment, the PKG modulators only modestly increase the activity of cGMP-dependent kinases but strongly increase the activity of cAMP-dependent protein kinase A (PKA), a close relative of PKG.

## 2. Results

### 2.1. Modulation of PKG Activity and Substrate Identification

To identify substrates for PKGs, the effect of modulators (ATP, cGMP, cAMP, PKG activator and PKG inhibitors) on recombinant PKGI and PKGII was tested in a high throughput setting, using PamChip^®^ 96 array plates. Each array comprises of 142 serine/threonine containing peptides which are derived from putative phosphorylation sites in the human phosphoproteome [19]. The peptide names consist of the protein they are derived from and the amino acid positions in that protein. First, incubations with or without ATP were performed to establish that the phosphorylation reaction for PKGI and PKGII is ATP-dependent (Appendix A). Next, a concentration series of recombinant PKGI and PKGII was tested on PamChip^®^ arrays to determine the desired PKG input for this study (Appendix A). We found that the assay is linear with enzyme input; however, at a high concentration, the linearity is lost. Based on the results, we chose 0.5 ng PKGI and 5 ng PKGII per array as protein input concentration in all the following experiments.

#### 2.1.1. Effect of cGMP and cAMP on PKG Activity

The phosphorylation activity of PKGI and PKGII on the peptides as a function of cGMP or cAMP concentration is shown in Figure 1. PKGI and PKGII show a big overlap in substrate preference, although there are also differences. The phosphorylation signal intensity of four peptides each for PKGI and PKGII (ERF_519_531 and VASP_232_244 as substrates for both PKGI and PKII, CFTR_761_773 and F263_454_466 as substrates for PKGI and CENPA_1_14 and H32_3_18 as substrates for PKGII) as a function of cGMP or cAMP concentration is shown in Figure 1. Signal intensity for PKGI and PKGII started to increase around 100 nM cGMP and 1 µM cAMP.

At saturating cGMP concentrations, the kinase activity increased about 30 times for PKGI and 16 times for PKGII. Higher concentrations of cAMP were required to achieve a stimulation of activity. Maximal activity for cAMP-activated PKGI was about half the value found with cGMP. For PKGII, the difference between the maximal activities with cGMP and cAMP was smaller (Figure 1). Variation of cGMP and cAMP concentrations allowed the determination of the activation constants (Ka) for both PKGI and PKGII (Table 1). Ka values obtained for PKGI (0.26 µM for cGMP and 22.4 µM for cAMP) were in good agreement with the data reported in the literature, but for PKGII, the measured Ka is higher than the reported values, which varied 20-fold for cGMP.

The response of PKGs to cGMP and cAMP also permits us to eliminate the effect of kinases that could be present as contaminants in the enzyme preparations. The activity of such kinases would not respond to the addition of these modulators. We excluded the peptides which already showed high signal intensity in the absence of cGMP or cAMP, with the signal either remaining constant or increasing further with elevated concentrations of cGMP or cAMP.

#### 2.1.2. Effect of PKG Activator and Inhibitors on PKG Activity

To further classify the substrate preferences of PKGI and PKGII, the effect of PKG activator 8-Br-cGMP, pan-PKG inhibitor Rp-8-pCPT-cGMPS (for both PKGI and PKGII) and the PKGI specific inhibitor Rp-8-Br-PET-cGMPS on the kinase activity was tested for three compound concentrations in the presence of cGMP. The modulation of PKGI or PKGII activity by these compounds is shown for the same four peptides (Figure 2).

The addition of PKG activator resulted in a concentration-dependent increase in phosphorylation (Figure 2a). The activator leads to a doubling of kinase activity for PKGI and on average a 50% increase for PKGII. The effect was most prominent for the peptide ERF_519_531 (an ETS domain-containing transcription factor ERF). The addition of PKGI-specific inhibitor Rp-8-Br-PET-cGMP resulted in the inhibition of PKGI kinase activity only, with no effect on PKGII activity (Figure 2b). On the other hand, the addition of pan-PKG inhibitor Rp-8-pCPT-cGMP resulted in a decrease in the kinase activity of both PKGI and PKGII for all four peptides (Figure 2c). These data show that the PKG activity can be modified by the addition of PKG activity modulators and this change can be read out by phosphorylation changes on the peptides of the peptide microarray.

#### 2.1.3. PKGI and PKGII Substrate Identification

First, peptides were selected for ATP-dependency—only those that showed a significant increase in signal upon addition of ATP at the highest concentrations of cGMP and cAMP were included in the analysis. Out of 142 peptides, 81 peptides showed statistically significant (*p* ≤ 0.05) ATP-dependent phosphorylation by PKGI and 92 by PKGII.

For substrate identification for the kinases, we devised a scoring system that included signal dependency on cGMP and cAMP concentration, phosphorylation induced by the PKG activator, and phosphorylation inhibited by PKG inhibitors (see Substrate Identification section of Materials and Methods and Appendix A for assignment of scores). For PKGI, data for the specific PKGI inhibitor, Rp-8-Br-PET-cGMPS were assessed and for PKGII, data for the pan-PKG inhibitor Rp-8-pCPT-cGMPs were used. The combined results obtained with the modulators amounted to a maximal value of ten and yielded a score indicating the preference of the kinases for each substrate. The substrates, their sequence, UniProt ID and scores for PKGI and PKGII are indicated in Table 2.

We checked the Human Protein Reference Database (www.hprd.org), UniProt Knowledge base (www.uniprot.org), and PhosphoSitePlus^®^ (www.phosphositeplus.org) for known PKG phosphorylation sites. Several peptides in Table 2 have already been reported as substrates for PKGI and/or PKGII such as VASP (S153, T278, S399), PDE5A_95_107 (cGMP-specific 3′, 5′-cyclic phosphodiesterase), RYR1_4317_4329 (ryanodine receptor 1), CFTR_761_773 (cystic fibrosis transmembrane conductance regulator), ERBB2_679_691 (receptor tyrosine kinase precursor), LIPS_944_956 (hormone-sensitive lipase) and BAD_69_81 (Bcl2 antagonist of cell death). In addition to these known substrates, novel PKGI and PKGII substrates were identified. PKGI and PKGII showed an overlap in substrate preference but also displayed differential preferences for substrates, as indicated by the scores. Figure 1 and Figure 2 show examples of good substrates for both PKGI and PKGII (ERF_519_531, VASP_232_244), substrates preferred by PKGI (CFTR_761_773, F263_454_466) and substrates preferred by PKGII (CENPA_1_14, H32_3_18).

The peptide sequences on PamChip^®^ arrays can be present in more than one protein. Selection of only the UniProt ID of named peptide leads to loss of relevant biological information. To identify proteins that correspond to the peptides, the peptide sequences were blasted against the UniProt database of human proteins (see Materials and Methods, Blast section). Only peptides with identical sequences (all amino acids identical) or similar sequences (having only conservative substitutions) were included (Table 3). Blasting yielded phosphosites with identical sequences in proteins such as the PKA subunits α and β proteins for KAPCG peptide, but also led to the identification of proteins with similar sequences, as for the PPR1A peptide. The threonine residue in this peptide was replaced by the serine residue in the protein phosphatase 1 regulatory subunit 1C.
ijms-22-01180-t003_Table 3Table 3Extended list of proteins phosphorylated by PKGI and or PKGII, (similarity = 1) after blasting of the peptides in Table 2.Peptides on STK PamChip^®^Hits after BlastingIDUniProt IDSequenceProtein NameUniProt IDSequenceKAPCG_192_206P22612VKGRTWTLCGTPEYLcAMP-dependent protein kinase catalytic subunit αcAMP-dependent protein kinase catalytic subunit βP17612P22694VKGRTWTLCGTPEYLVKGRTWTLCGTPEYLKPCB_19_31_A25SP05771RFARKGSLRQKNVProtein kinase C α typeP17252RFARKGSLRQKNVH32_3_18Q71DI3RTKQTARKSTGGKAPRHistone H3.1Histone H3.3Histone H3.3CHistone H3.tP68431P84243Q6NXT2Q16695RTKQTARKSTGGKAPRRTKQTARKSTGGKAPRRTKQTARKSTGGKAPRRTKQTARKSTGGKAPRRAF1_253_265P04049QRQRSTSTPNVHMSerine/Threonine-protein kinaseA-RafP10398QRIRSTSTPNVHMPPR1A_28_40Q13522QIRRRRPTPATLVProtein phosphatase 1 regulatory subunit 1CQ8WVI7QIRKRRPTPASLVNCF1_296_308P14598RGAPPRRSSIRNAPutative neutrophil cytosol factor 1CA8MVU1RGAPPRRSSIRNAADDB_706_718P35612KKKFRTPSFLKKSα-adducinP35611KKKFRTPSFLKKSRAP1B_172_184P61224PGKARKKSSCQLLRas-related protein Rap-1-b-like proteinA6NIZ1PGKARKKSSCQLLCREB1_126_138P16220EILSRRPSYRKILcAMP-responsive element modulatorQ03060EILSRRPSYRKILDESP_2842_2854P15924RSGSRRGSFDATGPlectinQ15149RAGSRRGSFDATG

Among the high scoring peptides identified, both PKGI and PKGII can phosphorylate the protein kinase PKCα and the regulatory subunit of protein kinase A (PKA). It was interesting to find that PKGII, but not PKGI, is able to phosphorylate the catalytic subunit of PKA, notably T198 in the PKA activation loop. This residue is conserved in the PKA α, β and γ catalytic subunits (www.phosphosite.org). PKGII is also able to phosphorylate histones which play a prominent role in DNA repair and chromosomal stability [27]. On the other hand, PKGI targets the serine/threonine-protein kinase A-Raf which regulates the mTOR signaling cascade, which is activated in tumors, insulin activation and in many other cellular processes [28]. mTOR (also known as FRAP) itself can be phosphorylated by both PKGI and PKGII on S2448, a site known to be phosphorylated by AKT1 and p70S6K (www.phosphosite.org).

As an additional check for the substrate quality, substrate motifs for PKGI and PKGII were determined (Appendix A). PKGI showed a preference for substrates containing an R/K-R/K-X-S/T- motif. The R/K-R/K-X-S/T motif is also common to other members of the AGC kinase family, such as PKA [19]. PKGII showed a preference for peptides with more positively charged amino acids at the N-terminal side, reflected in a G/R/K-X-K/G/R-X-R/K-R/K-X-S/T motif.

These experiments show that PKGI and PKGII respond differently to the addition of modulators with an increase or decrease in signal intensity. Furthermore, PKGI and PKGII have different substrate preference, which can be used to distinguish the two enzymes in a lysate, which is a complex mixture of many kinases. Knowledge of PKGI and PKGII substrates, in combination with their response to the modulators was used further to study the activity of PKGs in the retinal murine cell line 661W. 661W immortalized cone photoreceptor precursor cells are derived from the retinal tumor of a mouse expressing SV40 T antigen under the control of a photoreceptor specific promoter [29] and have been used as a cell model for retinal ciliopathies such as retinitis pigmentosa [17,18]. The role of PKGI and PKGII in retinal degeneration has not been elucidated yet.

### 2.2. Modulation of Kinase Activity in Retinal 661W Cells

#### 2.2.1. Effect of Modulators on Kinase Activity in 661W Cells

To study the effect of elevated cGMP on endogenous PKGI and PKGII activity and its downstream effects, we investigated PKG activity in the photoreceptor cell line, 661W. Cells were grown and lysed as described in Materials and Methods. The addition of modulators to a lysate ex vivo in an on-chip assay is expected to assess the effect on endogenous PKGI and PKGII activity and determine their contribution to kinase activity in the lysate.

We first compared the kinase profile of 661W cell lysate with those of the recombinant PKGs at 100 µM cGMP, as is visualized in a scatter plot (Figure 3a,b). The correlation coefficient between the kinase activity profile of the cells and recombinant PKGI and PKGII was 0.75 and 0.71, respectively. This is a relatively low correlation and it suggests a prominent role for kinases other than only the PKG family in the cells. 

The kinase activity in the cells was analyzed as function of the concentration of cGMP and cAMP, resulting in increased phosphorylation signal intensities on many peptides. The kinases in the lysate showed a stronger response to cAMP than to cGMP. Relative increases in phosphorylation signal intensities for three peptides with the highest signals in 661W cells at 1 µM concentration of cGMP or cAMP are shown in Figure 3c, as well as for VASP_150_162. The peptide VASP_150_162 is one of the three VASP peptides that is phosphorylated by PKGs and frequently used as a read out for PKG activity in studies on retinal degeneration [9]. The peptide phosphorylation increased at lower concentrations of cAMP than of cGMP. The Ka of cGMP in the 661W cell lysate was found to be around 10 µM for peptides that responded to these compounds, as is illustrated with the peptide VASP_150_162 in Figure 3d. For recombinant PKGI and PKGII, the Ka was 0.2 and 1.6 µM, respectively (Table 1). For cAMP, the Ka in cell lysate was around 0.1 μM (Figure 3e), whereas for recombinant PKGI and PKGII, Ka values of 7.6 and 39 µM, respectively, were found (Table 1). These results indicate that the kinase activity in the cell lysate was more responsive to cAMP than to cGMP. Addition of increasing concentrations of PKG activator resulted in increased kinase activity starting from 0.1 μM with a Ka of about 2.6 μM (Figure 3f). The value of PKG activator reported to activate PKGs is between 0.01 and 1 μM [20], much lower than the value found in the 661W cell lysate. The addition of PKG inhibitors (PKGI specific and pan–PKG) at concentrations where recombinant PKGI and/or PKGII activity was strongly inhibited did not lead to statistically significant inhibition of the kinase activity in the 661W lysate on any of the peptides on the array.

#### 2.2.2. Not PKG but PKA Is Modulated in 661W Cells

Since the concentrations of cGMP, cAMP, and PKG activator where modulation of kinase activity in 661W lysate is observed are not in line with the concentrations required to activate recombinant PKG and the inhibitors have no effect on signal intensity, this may indicate that a different (or additional) cyclic nucleotide binding kinase is activated in the cells. Therefore, to assess the type of kinases differentially activated in the cells, upstream kinase analysis was performed comparing cell lysate with and without cGMP or cAMP. In the upstream kinase analysis, the kinases most likely to be able to phosphorylate the peptide sequences on the array are identified (see Materials and Methods section). The kinases hypothesized to be activated in cGMP and cAMP condition in 661W cells in comparison with untreated control are shown in Figure 3g,h. In both analyses, Pim1 ranked highest. Although this kinase is able to phosphorylate many peptides, it does not respond to cyclic nucleotides, and therefore was excluded. Among the cyclic nucleotide binding kinases, the upstream kinase analysis suggested the kinases PKA, PKGI and PKGII as most likely affected in both cGMP and cAMP conditions. A possible activation of PKA is substantiated by the fact that the Ka values of cGMP, cAMP, and PKG activator obtained in 661W lysate match those reported for PKA activation [20].

To check the hypothesis that PKA rather than PKG is activated in 661W cells with the addition of PKG activity modulators, we first performed a substrate identification for the recombinant PKA catalytic subunit α on the STK PamChip^®^. This was done both in the presence and absence of PKA inhibitor peptide (PKAi). The substrates were identified on the basis of their activation and inhibition in the presence of ATP and PKAi, respectively. The set of PKA substrates showed a big overlap with the PKG substrates (Appendix A). All peptides shown in Figure 3c were found to be good PKA substrates, and also VASP_150_162. To investigate whether PKA is active in the 661W lysate, we added the PKAi to the cell lysate supplemented with cGMP or cAMP. The addition of PKAi to the lysate decreased phosphorylation of the PKA substrates when compared with the control (lysate without PKA inhibitor) (Figure 4a,b). However, the addition of PKAi did not result in significant inhibition of the signal intensity on peptides ERF_519_531 and H32_3_18, which are not PKA substrates.

We also checked the possibility that PKAi inhibits PKGI and PKGII activity by incubating the recombinant PKGs with 1 µM PKAi. Neither PKGI nor PKGII were inhibited by PKAi (Appendix A). We checked the effect of the modulators on the peptide VASP_232_244, ERF_519_531 and H32_3_18 that are good substrates for PKGI and or PKGII (Table 2). The peptide VASP_232_244 had no signal in 661W lysates, and addition of modulators did not have any effect. These data too are in agreement with a low activity of PKGI and PKGII in the 661W lysate. However, phosphorylation signals are present on the peptides ERF_519_531 and H32_3_18. Since PKG activity is low in the 661W lysate, the phosphorylation of these peptide must be due to activity of other kinases. S10 site of H32 is also known to be phosphorylated by a.o. AuroraA (www.uniprot.org).

The Ka values for cGMP, cAMP and 8-Br-cGMP and the lack of inhibition by PKG inhibitors, in addition to the strong inhibitory effect by PKAi, confirm that PKA is present at high concentrations in 661W cells.

## 3. Discussion

The genetic heterogeneity in the IRD group of diseases severely limits the development of mutation-specific treatments [2]. Therefore, it is imperative to identify targets where several disease-associated pathways convene for the design of treatments addressing an extended group of IRD patients. PKG has emerged as a promising target for the treatment of IRDs as its inhibition leads to photoreceptor preservation [8,9]. Therefore, identification of PKG substrates and their downstream signaling pathways in photoreceptors might help to find generic, new targets for the treatment of IRDs. More insight on the effect of PKG modulators on PKGs will not only be beneficial in IRDs but also for cancer research, as PKG activators have been shown to reduce cell proliferation, activate cell death and limit cell invasion in different cancer cell models [11,12,13,14,15,16].

Here, we identified novel substrates for PKGI and PKGII in a multiplex assay using a peptide microarray, which allowed investigation of the phosphorylation of the 142 peptides present on one array. The substrates for PKGI and PKGII were selected on the basis of a series of strict criteria: response to ATP, cGMP, cAMP, PKG activator and inhibitors, as assessed by statistical analysis, quality of the fit for cAMP and cGMP, and effect size and direction for the activators and inhibitors. Peptides were scored for each criterion. This approach resulted in the confirmation of several known PKG substrates such as VASP (S153, T278, S399), CREB, PDE5A, CFTR (see Table 2) for PKGI and PKGII and identification of novel substrates, that, to the best of our knowledge, have not been described before. We identified good substrates for PKGI and PKGII (e.g., VASP_232_244, ERF_519_531, GPR6_349_361) and substrates preferred by either one of the kinases (e.g., CFTR_761_773, F263_454_466 for PKGI, and H32_3_18 and RBL2_655_667 for PKGII) (see Table 2). Data obtained on the peptide micro-array confirmed the Ka values for cGMP and cAMP reported for the two enzymes. Comparison of PKG substrates with substrates for PKA revealed also differences in substrate preference between the PKG’s and PKA. The response to cyclic nucleotides, 8-Br-cGMP, and PKG inhibitors is also different for the PKGs and PKA [20]. Furthermore, we confirmed that PKAi is a nM inhibitor for PKA, and showed that it does not inhibit PKGI and PKGII at µM concentrations. These compounds, in combination with a peptide microarray were shown to be valuable tools to distinguish PKGI and PKGII from PKA activity in a complex environment like a cell lysate.

### Role of PKG Activation in Retinal Cells

In the 661W cell lysate, a multitude of kinases are present. Peptides on the microarray can be phosphorylated by many serine-threonine kinases, which makes elucidation of the role of PKG complicated. To overcome this limitation, we decided to make use of specific PKGI or PKGII modulators that can be added to the cell lysate, to activate or inhibit the protein kinase G family. In 661W retinal cell lysate, addition of both cGMP and cAMP resulted in an increase in the overall kinase activity (Figure 3c). As compared to studies using purified, recombinant PKGs, a higher cGMP concentration was required in the cell lysate to activate these kinases. The Ka of cGMP in the cell lysate was determined to be around 10 μM, which is close to the Ka for cGMP reported for the PKA family [20]. The Ka for cAMP in the cell lysate was reached at a much lower concentration level, i.e. 0.1 µM, but, again, this is also closer to the cAMP concentration required to activate PKA [20,21]. Activation by 8-Br-cGMP also occurred at a concentration more likely to activate PKA than PKGs [20]. Furthermore, we found that PKG inhibitors did not significantly change kinase activity of the lysates, whereas the addition of a PKA inhibitor resulted in a pronounced inhibitory effect of the kinase activity on the peptide micro array. From the recombinant PKGs study, the three VASP peptides on PamChip^®^ were found to be substrates for PKGI and PKGII. VASP_232_244, which is not phosphorylated by PKA but is phosphorylated by PKGs, had a very low signal intensity in the lysate. These observations suggest that with the addition of PKG modulators in the cell lysate, PKA activity is affected. These experimental data indicate that the main body of PKA in 661W lysate is present in an inactive form, and it becomes activated by the addition of kinase modulators. Gene expression studies show that a high concentration of PKA catalytic subunit α is present in 661W cells [17]. Our experimental data suggest that PKG is present at a low concentration in the cell lysate, because it is hardly activated or inhibited by these modulators, or at least is effectively shielded (and thus not exposed) to these modulators. Previous studies reported that both PKGI and PKGII are present in the 661W cells [30] and the gene expression analysis of the cone receptor cells has shown a high expression of PKGII [17]. The expression of PKA catalytic subunit β and PKGI is low in this cell line [17]. Our experiments indicate that PKG activity in 661W can only be modulated to a limited extent, whereas PKA activity responds highly to all PKG modulators tested. This may prompt us to reconsider the role of PKA in retinal degeneration and in future research investigate putative interactions between PKA and PKG.

PKG inhibitors and knockdown of PKG have been shown to delay retinal degeneration and therewith indicate a clear role for PKG. The role for PKG in retinal degeneration is based on its high affinity for cGMP. However, our data show that PKA can also be activated by cGMP, albeit at higher concentrations, in the range of those needed to open CNG channels [31]. Evidence for involvement of PKG comes from quite a number of studies. In RD mouse models, Paquet-Durand et al. showed, using immunofluorescence, co-localization of cGMP and PKG, inferring a role for PKG in retinal degeneration [8]. Phosphorylation of VASP at S238 is used as read out for PKG activity in several murine RP model-based studies [8,9].

However, our cell line studies revealed that the addition of PKG modulators can also affect the activity of PKA. Substrate identification showed that PKGII (Table 3), and not PKGI, is able to activate PKA catalytic subunits α, β, and γ by phosphorylating T198 in the activation loop, which is conserved in all three PKA subunits. It has also been shown that PKG phosphorylates PKA regulatory subunit I α, which activates PKA in eukaryotic cells [32]. The addition of PKAi to the cell lysate confirms the interrelationship between PKG and PKA and the key function of PKA. Therefore, the cross talk between PKG and PKA, i.e., PKG is activated by cGMP and activates PKA, might provide a plausible explanation for the predominant effect of PKA via the cGMP axis. In such an environment, the conventional route of PKA activation through the cAMP stimulus might be bypassed by the cGMP route. Based on our novel finding revealing the prominent PKA activity, the interplay of PKG and PKA axis in retina cells should be taken into account when studying the molecular events during retinal degeneration, and optimized conditions where PKA is not activated should be considered.

## 4. Materials and Methods

### 4.1. Materials

PKGIα (full length human recombinant protein type α) was obtained from Millipore and PKGII (full length human recombinant protein) was obtained from Thermo Fischer Scientific. PKA catalytic subunit alpha and PKA inhibitor peptide (PKAi) were from Merck. PKG activator (8-Br-cGMP) and PKG inhibitors (Rp-8-Br-PET-cGMPS, Rp-8-pCPT-cGMPS) were provided by BIOLOG Life Science Institute (Bremen, Germany). cGMP and cAMP sodium salts were purchased from Sigma-Aldrich. The immortalized murine retinal photoreceptor precursor cells 661W [31,32] were generously provided by Dr. Muayyad Al-Ubaidi (University of Houston, Houston, Texas, USA). Dulbecco’s modified Eagle medium (DMEM), fetal bovine serum (FBS), penicillin, and streptomycin were purchased from Gibco. Mammalian protein extraction reagent (M-PER^TM^) buffer, Halt^TM^ protease and phosphatase inhibitor cocktails and the Coomassie Plus (Bradford) assay kit were purchased from Thermo Fischer Scientific.

### 4.2. Cell Culture and Lysis Procedure

661W cells were cultured in DMEM supplemented with 10% FBS, 100 U/mL penicillin, and 100 μg/mL streptomycin. The cells were maintained at 37 °C in a humidified atmosphere of 5% CO_2_. At 80% confluency (passage number 19), the medium was removed and cells were washed with ice cold PBS. The cells were lysed with lysis buffer (MPER with 1:100 phosphatase inhibitor cocktail and protease inhibitor cocktail) for 15 min on ice. The lysate was centrifuged at 16,000× *g* for 15 min at 4 °C. The supernatant was immediately aliquoted, flash-frozen, and stored at −80 °C. The protein content of the lysate was determined using the Bradford Protein Assay [33].

### 4.3. Kinase Activity Measurements

Kinase activity of recombinant kinases and cell lysates was determined on STK PamChip^®^ arrays (product # 87,102), each comprising 142 peptides derived from human proteins, according to the instructions of the manufacturer (PamGene International B.V., ‘s-Hertogenbosch, North Brabant, The Netherlands). An antibody mix detects the phosphorylated Ser/Thr amino acid residues, which is confirmed by addition of FITC-conjugated secondary antibody [19].

The assay mix consisted of protein kinase buffer (PamGene International BV. ‘s-Hertogenbosch, North Brabant, The Netherlands), 0.01% BSA, STK primary antibody mix and recombinant protein or lysate. The protein amount used in the assays was 0.5 ng/array for PKGI, 5 ng/array for PKGII, 1 ng/array for PKA and 0.5 µg/array for 661W cell lysate, unless indicated otherwise. The cyclic nucleotide concentration range varied from 3.3 nM to 1 mM for cGMP or to 3.3 mM for cAMP. For PKG activator or inhibitors, a concentration range from 0.025 to 2.5 μM was used [20]. The effect of PKG activator or inhibitors was tested in the presence of 0.2 μM cGMP. In all experiments, 400 µM ATP was present. A total assay volume of 40 μL was applied per array.

### 4.4. Instrumentation

All experiments were performed in triplicate on a PamStation^®^ on which up to 96 assays can be performed simultaneously (PamGene International B.V., ‘s-Hertogenbosch, North Brabant, The Netherlands). To prevent aspecific binding to the arrays, the PamChips^®^ were blocked with 2% BSA by pumping it up and down 30 times through the array. The chips were then washed 3× with Protein Kinase buffer and assay mix was applied. The assay mix was pumped up and down the arrays for 60 min. The arrays were washed 3× and the detection mix comprising of FITC labeled secondary antibody was applied on the PamChips^®^. The signals were recorded at multiple exposure times by a CCD camera [34].

### 4.5. Data Analysis

Signals on all peptides at all exposure times were quantified by BioNavigator^®^ software version 6.3.67.0 (PamGene International B.V., ‘s-Hertogenbosch, North Brabant, The Netherlands). For each peptide on the array, the software calculates a single value for images obtained at multiple exposure times (exposure time scaling) [34]. Statistical methods such as *T*-tests and one-way ANOVA were used to compare two or more groups. Bionavigator^®^ software was used for data visualization and statistical analysis. The graphs were made with GraphPad Prism 9 software (GraphPad Software, San Diego, CA, USA).

To identify kinases that are able to phosphorylate the peptide sequences on the array, upstream kinase analysis was performed [34]. The phosphorylation changes between two groups were compared and linked to kinases known to phosphorylate these sites (upstream kinases). The knowledge base is compiled from experimental data from several databases and theoretical interactions (PhosphoNet). The result provides a list of kinases that might be differentially active.

### 4.6. Blast

Peptide sequences were blasted in UniProt against human proteins using the PAM30 substitution matrix [35]. Similarity was defined as the ratio of the PAM30 score for a retrieved sequence and the original sequence. Peptides with a similarity score equal to 1 were included.

### 4.7. Substrate Identification

Results obtained in experiments with the different modulators were assessed using a scoring system for each experiment. The 142 peptides were first scored on the basis of statistically significant (*p* ≤ 0.05) ATP-dependent phosphorylation by PKGI or PKGII in the presence of cGMP. In the next step, the peptides were scored for a statistically significant increase in phosphorylation by PKGI or PKGII with increasing cGMP or cAMP concentration (*p* ≤ 0.05 between lowest and highest concentrations). The quality of the fit (R^2^ > 0.8) for cAMP and cGMP for each peptide was also included in the scoring. Next, the peptides were scored for statistically significant activation or inhibition in response to PKG activator or PKG inhibitors, respectively. Noise on signals was eliminated by checking for the expected direction of change and size of the fold change. To identify the best substrates for PKGI and PKGII, all scores were combined in one value. Based on the score, the peptides were classified as good, intermediate, or poor substrates for PKGI and or PKGII (see Appendix A).

### 4.8. Substrate Motifs

Seq2Logo 2.0 was used to generate substrate motifs based on the sequences listed in Table 2. For this purpose, the peptide sequences from Table 2 were aligned relative to the target Ser residue and only residues from position −5 to +5 were considered. To account for differences in signal intensity, a weight corresponding to the relative signal intensity was applied when entering sequences into the program.

## 5. Conclusions

PKG has emerged as a crucial target to design treatment for a highly heterogeneous group of IRDs. Here, we used peptide microarrays with PKG modulators for high throughput substrate identification of PKGI and PKGII. We were able to determine substrates specific for each kinase, and found a large overlap between substrates for both PKGs. We also showed that these modulators stimulate PKA activity in retinal cells.

## Figures and Tables

**Figure 1 ijms-22-01180-f001:**
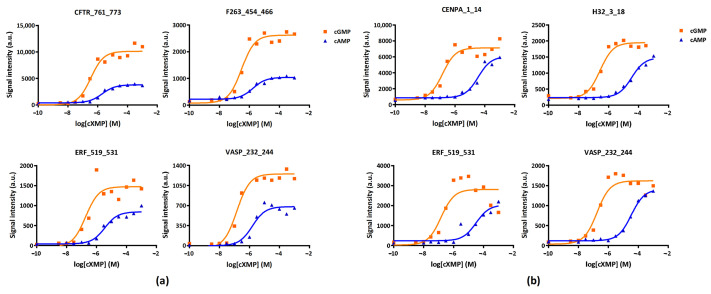
Effect of cGMP and cAMP concentrations on signal intensity of (**a**) PKGI or (**b**) PKGII on selected peptides. The signal intensity is the mean of triplicate measurements.

**Figure 2 ijms-22-01180-f002:**
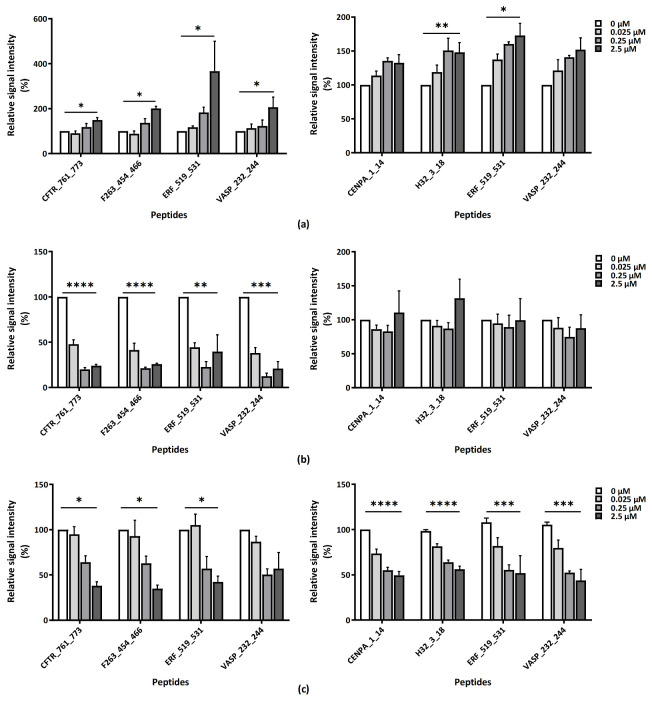
The effect of modulators on the activity of PKGI (left) and PKGII (right) for selected peptides: (**a**) PKG activator (8-Br-cGMP), (**b**) PKGI Inhibitor (Rp-8-Br-PET-cGMPS), and (**c**) pan-PKG inhibitor (Rp-8-pCPT-cGMPS). The concentration of the modulators was varied as indicated in the legend in the presence of 0.2 µM cGMP. Relative signal intensity was measured in triplicate and expressed with respect to the condition without the modulator. The significant changes with modulators were determined with one-way ANOVA with significance indicated as * (*p* ≤ 0.05), ** (*p* ≤ 0.01), *** (*p* ≤ 0.001) or **** [(*p* ≤ 0.0001)]**.**

**Figure 3 ijms-22-01180-f003:**
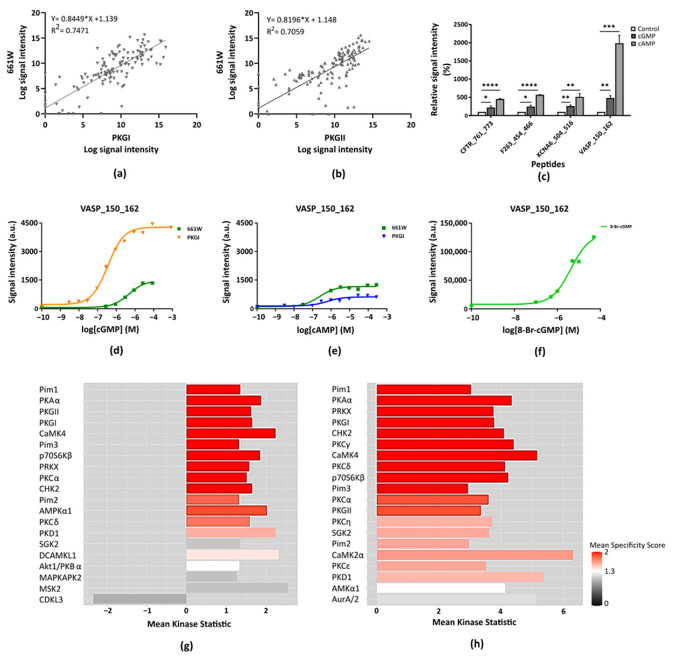
(**a**,**b**) Scatter plot of signal intensities of 661W cell lysate and the recombinant kinases PKGI (**a**) and PKGII (**b**) at 100 µM cGMP. (**c**) Modulation of kinase activity by 1 µM cGMP and cAMP in 661W cell lysate for selected peptides. Relative signal intensity was measured in triplicate and expressed with respect to the condition without the modulator. The significance of changes with cGMP and cAMPs was determined by Unpaired *T*-test with significance indicated as * (*p* ≤ 0.05), ** (*p* ≤ 0.01) and *** (*p* ≤ 0.001), **** [(*p* ≤ 0.0001)]. (**d**,**e**) Modulation of phosphorylation of the peptide VASP_150_162 for recombinant PKGI and 661W cell lysate with an increase in cGMP (**d**) and cAMP (**e**) concentrations. The relative signal intensity of VASP_150_162 is plotted against log concentration of cGMP (left) or cAMP (right). Data points are the mean of three replicates. (**f**) Modulation of kinase activity for peptide VASP_150_162 in 661W cell lysate with increasing concentrations of PKG Activator 8-Br-cGMP (n = 1). (**g**,**h**) Kinases predicted by upstream kinase analysis to be activated in 661Wcells at 1 µM cGMP (**g**) or cAMP (**h**).

**Figure 4 ijms-22-01180-f004:**
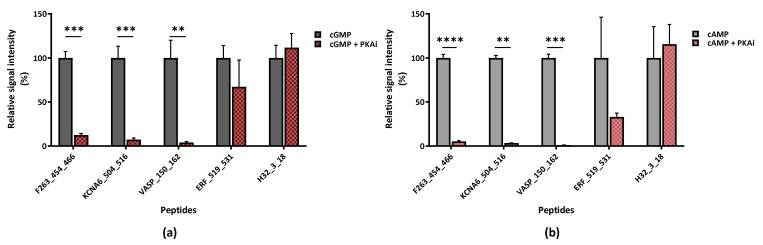
Phosphorylation of selected peptides by 661W cell lysate in the presence of either (**a**) cGMP (10 µM) or (**b**) cAMP (0.1 µM) with and without PKAi (1 µM). The significance of changes in cGMP and cAMPs was determined by unpaired T-tests with significance indicated as ** (*p* ≤ 0.01), *** (*p* ≤ 0.001) and **** [(*p* ≤ 0.0001)]**.**

**Table 1 ijms-22-01180-t001:** Comparison of experimentally determined Ka values of cGMP and cAMP for PKGI and PKGII with literature values.

cXMP	PKGI Ka (µM)		PKGII Ka (µM)	
	Measured	Literature	Measured	Literature
cGMP	0.26	0.1–0.2 [20,21,22,23,24]	1.6	0.04–0.8 [20,24,25,26]
cAMP	22.4	7.6–39 [20,21]	27	~12 [20]

**Table 2 ijms-22-01180-t002:** Peptide substrates for PKGI and PKGII, their sequence, protein name, UniProtID and score. Scores for PKGI and PKGII range from 1 to 10. For sites already known to be phosphorylated by PKGI or PKGII, the database containing this information is added. A—PhosphoSitePlus^®^, B—Human Protein Reference Database, C—UniProt. Color scheme according to the score—10–8: Good, 7–4: Intermediate and 3–0: Poor substrate.

Peptide ID	UniProt ID	Peptide Sequence	Description	PKGI Score	PKGII Score	Ref.
ERF_519_531	P50548	GEAGGPLTPRRVS	ETS domain-containing transcription factor ERF	10	9	
VASP_232_244	P50552	GAKLRKVSKQEEA	Vasodilator-stimulated phosphoprotein	10	8	A (PKGI)
CREB1_126_138	P16220	EILSRRPSYRKIL	cAMP response element-binding protein	10	6	A, B, C (PKGI)
CSF1R_701_713	P07333	NIHLEKKYVRRDS	Macrophage colony-stimulating factor 1 receptor precursor	10	7	
EPB42_241_253	P16452	LLNKRRGSVPILR	Erythrocyte membrane protein band 4.2	9	7	
GBRB2_427_439	P47870	SRLRRRASQLKIT	Gamma-aminobutyric acid receptor subunit beta-2 precursor	10	6	
GPSM2_394_406	P81274	PKLGRRHSMENME	G-protein-signaling modulator 2	10	7	
GRIK2_708_720	Q13002	FMSSRRQSVLVKS	Glutamate receptor, ionotropic kainate 2 precursor	10	4	
PDE5A_95_107	Q76074	GTPTRKISASEFD	cGMP-specific 3’,5’-cyclic phosphodiesterase	10	6	
PTN12_32_44	Q05209	FMRLRRLSTKYRT	Tyrosine-protein phosphatase non-receptor type 12	10	5	
RS6_228_240	P62753	IAKRRRLSSLRAS	40S ribosomal protein S6	10	5	
RYR1_4317_4329	P21817	VRRLRRLTAREAA	Ryanodine receptor 1	9	5	
VTNC_390_402	P04004	NQNSRRPSRATWL	Vitronectin precursor	10	4	
CFTR_761_773	P13569	LQARRRQSVLNLM	Cystic fibrosis transmembrane conductance regulator	10	3	A, B (PKGI)
F263_454_466	Q16875	NPLMRRNSVTPLA	6-phosphofructo-2-kinase/fructose-2,6-biphosphatase 3	10	3	
KPB1_1011_1023	P46020	QVEFRRLSISAES	Phosphorylase b kinase regulatory subunit alpha, skeletal muscle isoform	9	3	
MYPC3_268_280	Q14896	LSAFRRTSLAGGG	Myosin-binding protein C, cardiac-type	10	4	
TY3H_65_77	P07101	FIGRRQSLIEDAR	Tyrosine 3-monooxygenase	9	3	
VASP_271_283	P50552	LARRRKATQVGEK	Vasodilator-stimulated phosphoprotein	8	3	A (PKGI)
ANXA1_209_221	P04083	AGERRKGTDVNVF	Annexin A1	7	9	
GPR6_349_361	P46095	QSKVPFRSRSPSE	Sphingosine 1-phosphate receptor GPR6	9	8	
KIF2C_105_118_S106G	Q99661	EGLRSRSTRMSTVS	Kinesin-like protein KIF2C	9	7	
ADDB_706_718	P35612	KKKFRTPSFLKKS	Beta-adducin	8	7	
CAC1C_1974_1986	Q13936	ASLGRRASFHLEC	Voltage-dependent L-type calcium channel subunit α-1C	9	7	
KAP2_92_104	P13861	SRFNRRVSVCAET	cAMP-dependent protein kinase type II-alpha regulatory subunit	6	5	
KCNA2_442_454	P16389	PDLKKSRSASTIS	Potassium voltage-gated channel subfamily A member 2	9	7	
NCF1_296_308	P14598	RGAPPRRSSIRNA	Neutrophil cytosol factor 1	9	7	
MPIP1_172_184	P30304	FTQRQNSAPARML	M-phase inducer phosphatase 1	9	4	
ART_025_ CXGLRRWSLGGLRRWSL	Na	GLRRWSLGGLRRWSL	Peptide based on kemptide sequence	8	4	
CDN1A_139_151	P38936	GRKRRQTSMTDFY	Cyclin-dependent kinase inhibitor 1	6	3	
KCNA6_504_516	P17658	ANRERRPSYLPTP	Potassium voltage-gated channel subfamily A member 6	9	3	
ERBB2_679_691	P04626	QQKIRKYTMRRLL	Receptor tyrosine-protein kinase erbB-2 precursor	7	8	A (PKGII)
DESP_2842_2854	P15924	RSGSRRGSFDATG	Desmoplakin	8	7	
KCNA3_461_473	P22001	EELRKARSNSTLS	Potassium voltage-gated channel subfamily A member 3	8	7	
RAF1_253_265	P04049	QRQRSTSTPNVHM	RAF proto-oncogene serine/threonine-protein kinase	8	7	
RAP1B_172_184	P61224	PGKARKKSSCQLL	Ras-related protein Rap-1b precursor	7	7	
KAP3_107_119	P31323	NRFTRRASVCAEA	cAMP-dependent protein kinase type II-beta regulatory subunit	8	4	
TOP2A_1463_1475	P11388	RRKRKPSTSDDSD	DNA topoisomerase 2-alpha	6	5	
ADRB2_338_350	P07550	ELLCLRRSSLKAY	Beta-2 adrenergic receptor	8	4	
ANDR_785_797	P10275	VRMRHLSQEFGWL	Androgen receptor	6	4	
REL_260_272	Q04864	KMQLRRPSDQEVS	C-Rel proto-oncogene protein	6	4	
VASP_150_162	P50552	EHIERRVSNAGGP	Vasodilator-stimulated phosphoprotein	8	4	A (PKGI)
PTK6_436_448	Q13882	ALRERLSSFTSYE	Tyrosine-protein kinase 6	7	1	
KPCB_19_31_A25S	P05771	RFARKGSLRQKNV	Protein kinase C β	5	8	
PLM_76_88	O00168	EEGTFRSSIRRLS	Phospholemman precursor	7	7	
FRAP_2443_2455	P42345	RTRTDSYSAGQSV	FKBP12-rapamycin complex-associated protein (mTOR)	7	7	
LIPS_944_956	Q05469	GFHPRRSSQGATQ	Hormone-sensitive lipase	7	7	A (PKGI)
PPR1A_28_40	Q13522	QIRRRRPTPATLV	Protein phosphatase 1 regulatory subunit 1A	4	7	
GYS2_1_13	P54840	MLRGRSLSVTSLG	Glycogen synthase, liver	5	4	
STK6_283_295	O14965	SSRRTTLCGTLDY	Serine/threonine-protein kinase 6 (Aurora A)	5	3	
PDPK1_27_39	O15530	SMVRTQTESSTPP	3-phosphoinositide-dependent protein kinase 1	3	9	
BAD_69_81	Q92934	IRSRHSSYPAGTE	Bcl2 antagonist of cell death	5	8	A(PKGI)
H2B1B_ 27_40	P33778	GKKRKRSRKESYSI	Histone H2B type 1-B	3	8	
NMDZ1_890_902	Q05586	SFKRRRSSKDTST	Glutamate [NMDA] receptor subunit zeta-1 precursor	4	8	
NOS3_1171_1183	P29474	SRIRTQSFSLQER	Nitric oxide synthase, endothelial	6	8	A (PKGI)
PLEK_106_118	P08567	GQKFARKSTRRSI	Pleckstrin	4	8	
H32_3_18	Q71DI3	RTKQTARKSTGGKAPR	Histone H3.2	4	9	
CENPA_1_14	P49450	MGPRRRSRKPEAPR	Histone H3-like centromeric protein A	3	7	
RBL2_655_667	Q08999	GLGRSITSPTTLY	Retinoblastoma-like protein 2	1	8	
KAPCG_192_206	P22612	VKGRTWTLCGTPEYL	cAMP-dependent protein kinase catalytic subunit γ	0	7

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
