# Peer review of "Identification of Novel Substrates for cGMP Dependent Protein Kinase (PKG) through Kinase Activity Profiling to Understand Its Putative Role in Inherited Retinal Degeneration"

_ijms, 2021, doi:10.3390/ijms22031180_

Round 1

Reviewer 1 Report

The current paper is an interesting contribution to the field of inherited retinal degeneration (IRD) as it sheds more insights on the cGMP-dependent protein kinase (PKG). The methodology used is sound, and the results support the conclusions. The plagiarism check did not show any overlap with previously published data.

The statistical analysis part needs improvement. Specifically, parametric tests (ANOVA and t-test) are not adequate since their data does not follow a normal distribution. Therefore, the authors are encouraged to replace ANOVA with the Kruskal-Wallis test and the t-test with the Mann-Whitney U test.

The limitations section is lacking. Specifically, the authors have used a murine cell line (661W) as a model to identify the substrates of PKGI and PKGII. It is well known that cell lines (even human ones) are not the best suited for studying the molecular mechanisms of IRDs. Furthermore, even fibroblasts derived from patients were not also giving accurate results… This idea needs to be mentioned and discussed…

Many meaningful sentences throughout the manuscript lack references, e.g., lines 44-52, lines 303- 306…The authors are encouraged to add adequate references.

Author Response

Dear Ms. Liu

Thank you for providing valuable comments of the reviewers which has helped us to improve the clarity of the manuscript. We also rephrased a few sentences and corrected some typos.

Response to comments by reviewer 1-

Comment 1: The statistical analysis part needs improvement. Specifically, parametric tests (ANOVA and t-test) are not adequate since their data does not follow a normal distribution. Therefore, the authors are encouraged to replace ANOVA with the Kruskal-Wallis test and the t-test with the Mann-Whitney U test.

Response: The authors thank the reviewer for this relevant suggestion. The reviewer is correct, but we do not think that our current analysis would benefit a lot from using non-parametric statistical methods instead of parametric methods. If clear deviations from normality occur in the data, this is mostly in the form of tails/outliers. If this occurs, the parametric methods may be somewhat conservative when selecting peptides as substrates. The peptides in the tails with low signal intensities and the outliers will have a low score and be considered less interesting PKG substrates. In general, we think the currently used statistical methods are adequate for the purpose of providing a consistent manner of assigning peptides as substrates for PKG kinases.

Comment 2: The limitations section is lacking. Specifically, the authors have used a murine cell line (661W) as a model to identify the substrates of PKGI and PKGII. It is well known that cell lines (even human ones) are not the best suited for studying the molecular mechanisms of IRDs. Furthermore, even fibroblasts derived from patients were not also giving accurate results… This idea needs to be mentioned and discussed…

Response: The authors thank the reviewer for this comment. The 661W cells are used to study molecular mechanisms of IRDs and in this study, we used these cells also because they express both isoforms of PKG i.e. PKGI and PKGII. We have mentioned this in the text (page 11, lines 256-258). In the first part of this study with the recombinant PKGs, we showed that the PKG activator and PKG inhibitors are able to modulate activity of PKGI and/or PKGII. Subsequently, using the 661W cells, we tested the effect of these modulators on cell lysate, to specifically activate PKGI or PKGII and to assess their activity in a lysate that contains many kinases that do not respond to the modulators. In a follow up study, we intend to stress the cells with specific compounds to mimic IRDs and investigate the effect on PKG activity.

Comment 3: Many meaningful sentences throughout the manuscript lack references, e.g., lines 44-52 (updated page 2, lines 49-59), lines 303- 306 (updated page 14, lines 368)…The authors are encouraged to add adequate references.

Response: We have added the references to the above-mentioned lines and also to page 11, line 271 and page 16, line 525.

Reviewer 2 Report

The manuscript by Roy et al describe their search and identification of peptides that can be phosphorylated by PKGI and PKGII based on the idea that PKG inhibition has been shown to slow down retinal degeneration in mouse models of some retinal degenerative diseases. Thus, PKG sites could potentially be therapeutic targets for IRDs. They start with a high throughput screen from human peptides, which do not appear to be eye-specific. They identify peptides and show some are better phosphorylated by PKGI or PKGII or both for 6 of the 142 peptides screened. Then they appear to go back to the their library of peptides to come up with PKGI and PKGII scores for each (Table 2) and then blast a subset of these peptides to see if they occur in other proteins (Table 3). To give the authors some aspect of vision, they assayed phosphorylation of these peptides in the presence of extracts from 661W cells, which are immortalized cells derived from retinal tumor cells that express components consistent with cone photoreceptors. Then they diverge and veer off into examining PKA.

Overall, the findings are good. It is a little hard to read because of the different peptides used to illustrate their activity are numerous and appear and disappear from figure to figure. The idea of identifying new targets for inherited retinal degeneration diseases became a little lost in the more general search and identification of possible PKG substrates; however, it is a good study to begin such studies.

Here are my specific comments:

  1. In the introduction, page 2, line 47, please reread what the sodium/potassium/calcium exchanger does (hint, it’s called an exchanger for a reason) and fix it in the text.
  2. Why were the 6 peptides in Figure 1 selected to illustrate cyclic nucleotide and PKG dependence? They do not appear to be important in the 661W extracts (except maybe H32_3_18?). Would that not suggest they may not have significance in vision?
  3. In Figure 2, do the authors think the PKGII inhibitor is too non-specific or is there another explanation? The authors show the data, make the statement, and leave the reader hanging. Perhaps, an extra sentence or two – even if it’s speculative – are needed if they want to show this data.
  4. Why is Table 3 a blast result of a subset of Table 2? Were these peptide specifically selected based on their PKG scores? Were all the peptides blasted, and the authors decided to show ones that were of interest?
  5. The labels in Figure 3b are too small.
  6. Figure 4: I’m not sure why the authors say that H32 peptide phosphorylation “must be due to activity of other kinases”. In Figures 1 and 2, it appears H32_3_18 is PKGII-dependent and the PKAi has no effect suggesting that unlike the other peptides in Figure 4, is independent of PKA. What did I miss?

Author Response

Dear Ms. Liu

Thank you for providing valuable comments of the reviewers which have helped us to improve the clarity of the manuscript. We also rephrased a few sentences and corrected some typos.

Response to comments by reviewer 2-

Comment 1: In the introduction, page 2, line 47 (updated page 2, line 52), please reread what the sodium/potassium/calcium exchanger does (hint, it’s called an exchanger for a reason) and fix it in the text.

Response: Following the suggestions of the reviewer we have modified the section describing the role of Na+/K+/Ca2+ exchanger in phototransduction (page2, lines 52-54).

Comment 2: Why were the 6 peptides in Figure 1 selected to illustrate cyclic nucleotide and PKG dependence? They do not appear to be important in the 661W extracts (except maybe H32_3_18?). Would that not suggest they may not have significance in vision?

Response: In Fig. 1-2, four peptides are selected each for recombinant PKGI and PKGII (ERF_519_531 and VASP_232_244 as substrates for both PKGI and PKII, CFTR_761_773 and F263_454_466 as substrates for PKGI and CENPA_1_14 and H32_3_18 as substrates for PKGII). The selection was based on the peptide score in Table 2. To illustrate the effect of cAMP and cGMP in cell lysates in fig.3c, we now make clear that we selected the three peptides with the highest phosphorylation signals, and we added data for the peptide CFTR_761_773 (second in signal intensity). The text in the manuscript has been adjusted to clarify this (page 11, lines- 266-271). The peptide H32_3_18 had a low signal intensity in the cell lysate and its signal intensity is not affected in any condition. This supports the conclusion that PKG activity is very low in the lysate. Since we used non-stressed 661W cells, it is too early to suggest that the peptide H32_3_18 might not have significance in vision. We will extend this study to retina tissue and based on those results we will be able to formulate conclusive statements regarding this peptide.

Comment 3: In Figure 2, do the authors think the PKGII inhibitor is too non-specific or is there another explanation? The authors show the data, make the statement, and leave the reader hanging. Perhaps, an extra sentence or two – even if it’s speculative – are needed if they want to show this data.

Response: In our study, we use two PKG inhibitors, the PKGI specific inhibitor Rp-8-Br-PET-cGMPS and the PKGs specific inhibitor Rp-8-pCPT-cGMPS. We have changed the name for the latter to ‘pan-PKG inhibitor.’ For the pan-PKG inhibitor, we do see it inhibiting the activity of both PKGI and PKGII while the PKGI specific inhibitor inhibits PKGI only (Fig.2, b-c). As suggested, we have included a sentence explaining the outcome from the recombinant PKGI and PKGII study. Specifically, we explained that the PKG activity can be modulated with addition of the PKG modulators and that we are able to measure the resulting change in the phosphorylation of peptides on the Pamchip® array (page 4, lines 156-158).

Comment 4: Why is Table 3 a blast result of a subset of Table 2? Were these peptide specifically selected based on their PKG scores? Were all the peptides blasted, and the authors decided to show ones that were of interest?

Response: All peptides in Table 2 have been blasted against the UniProt database. Table 3 shows only those the proteins that contain an identical or similar amino acid sequence with the same number of amino acids as the original sequence (i.e. similarity=1). The Materials and Methods part describing the blast method (page 16, line 525 and the results section (page 9, lines 201-210) have been extended to clarify this point. 

Comment 5: The labels in Figure 3b are too small.

Response: We have increased the size of legends in Fig. 3.

Comment 6: Figure 4: I’m not sure why the authors say that H32 peptide phosphorylation “must be due to activity of other kinases”. In Figures 1 and 2, it appears H32_3_18 is PKGII-dependent and the PKAi has no effect suggesting that unlike the other peptides in Figure 4, is independent of PKA. What did I miss?

Response: We regret that this sentence was not clear and have adjusted the text in the article (page 13, lines- 352-359). We added subheadings in section 2.2 to emphasize that PKG seems not affected by modulators in 661W lysate, but that most likely PKA activity is affected.

Reviewer 3 Report

The manuscript entitled “Identification of novel substrates for cGMP dependent protein 3 kinase (PKG) through kinase activity profiling to understand 4 its putative role in inherited retinal degeneration” deals with the use of human recombinant PKGs in combination with PKG activity modulators (cGMP, 17 3’,5’-cyclic adenosine monophosphate (cAMP), PKG activator, and PKG inhibitors) on a multiplex peptide microarray to identify substrates for PKGI and PKGII.  The authors also applied the developed technology in combination with PKG modulators to monitor kinase activity in the retinal cell line 661W which is used as a model system for Inherited retinal degenerative diseases.

The techniques used in the present manuscript are well suited and adequate for its goal.

The manuscript is well written and easy to understand. The references used in the manuscript are recent and are suitable.  As far as I am concerned, this is the first time the activity of PKGs in the retinal murine cell line 661W is studied ex vivo with the use of PKG modulators.

In my opinion, the findings are interesting for a broader community and deserve to be published. Despite its great potential, the paper comes some minor issues which are addressed below:

  • Figure 3e axis and legend are barely readable, please increase size.
  • In the references section there is a lack of consistency in the format of the referenced journals, some are in normal some in italics, some have iso name while others have the full journal name. Please use the same style in all the reference for the sake of coherence.

Author Response

Dear Ms. Liu

Thank you for providing valuable comments of the reviewers which have helped us to improve the clarity of the manuscript. We also rephrased a few sentences and corrected some typos.

Comment 1:  Figure 3e axis and legend are barely readable, please increase size.

Response: We have increased the size of the legends for Fig. 3.

Comment 2: In the references section there is a lack of consistency in the format of the referenced journals, some are in normal some in italics, some have iso name while others have the full journal name. Please use the same style in all the reference for the sake of coherence.

Response: The authors appreciate the reviewer for detecting this detail. We now have a consistent APA style for all the references.